# RTMPose: Real-Time Models for Multi-Person Pose Estimation

## Abstract

Recent studies on 2D pose estimation have achieved excellent performance on public benchmarks, yet its application in the industrial community still suffers from heavy model parameters and high latency. To bridge this gap, we systematically explore key factors in pose estimation including paradigm, model architecture, training strategy, and deployment, and present a high-performance real-time multi-person pose estimation pipeline, **RTMPose**. Our RTMPose-m achieves **75.8% AP on COCO** with **90+ FPS** on an Intel i7-11700 CPU and **430+ FPS** on an NVIDIA GTX 1660 Ti GPU, and RTMPose-x achieves **65.3% AP on COCO-WholeBody**. We further test RTMPose on mobile devices to evaluate its potential in critical real-time applications. RTMPose-s model achieves **72.2% AP on COCO** with **70+ FPS** on a Snapdragon 865 chip, outperforming existing methods used by industrial companies.

## 1 Introduction

Real-time human pose estimation is appealing to various applications such as human-computer interaction, action recognition, sports analysis, and VTuber techniques. Despite the stunning progress (Sun et al., 2019; Xu et al., 2022b) on academic benchmarks (Lin et al., 2014; **?**), it remains a challenging task to perform robust and real-time multi-person pose estimation on devices with limited computing power. Recent attempts narrow the gap with efficient network architecture (Yu et al., 2021a; Bazarevsky et al., 2020; Votel et al., 2023) and detection-free paradigms (Kreiss et al., 2019; Geng et al., 2021; Shi et al., 2022), which is yet inadequate to reach satisfactory performance for industrial applications.

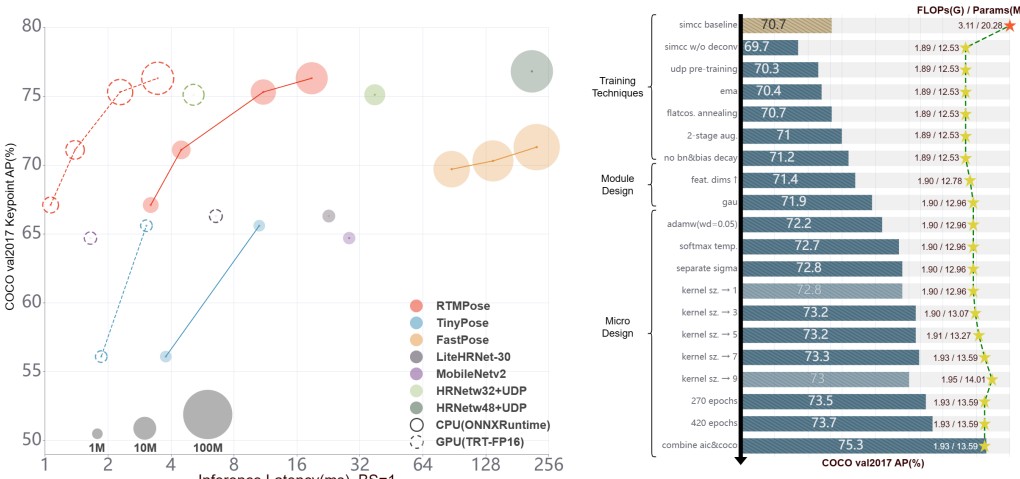

Figure 1: Left: Comparison of RTMPose and open-source libraries on COCO val set regarding model size, latency, and precision. The circle size represents the relative size of model parameters; Right: Step-by-step improvements from a SimCC baseline.

In this work, we systematically study key factors that affect the performance and latency of 2D multi-person pose estimation pipelines from five aspects: paradigm, backbone network, localization method, training strategy, and deployment. With a collection of optimizations, we introduce **RTMPose**, a new series of **R**eal-**T**ime **M**odels for **Pose** estimation.

First, RTMPose employs a top-down approach by using an off-the-shelf detector to obtain bounding boxes and then estimating the pose of each person individually. Top-down algorithms have been stereotyped as accurate but slow, due to the extra detection process and increasing workload in crowd scenes. However, benefiting from the excellent efficiency of real-time detectors (RangiLyu, 2021; Lyu et al., 2022), the detection part is no longer a bottleneck of the inference speed of top-down methods. In most scenarios (within 6 persons per image), the proposed lightweight pose estimation network is able to perform multiple forward passes for all instances in real time.

Second, RTMPose adopts CSPNeXt (Lyu et al., 2022) as the backbone, which is first designed for object detection. Backbones designed for image classification (He et al., 2015; Sandler et al., 2018) are suboptimal for dense prediction tasks like object detection, pose estimation and semantic segmentation, etc. Some backbones leveraging high-resolution feature maps (Sun et al., 2019; Yu et al., 2021a) or advanced transformer architectures (Dosovitskiy et al., 2021) achieve high accuracy on public pose estimation benchmarks, but suffer from high computational cost, high inference latency, or difficulties in deployment. CSPNeXt shows a good balance of speed and accuracy and is deployment-friendly.

Third, RTMPose predicts keypoints using a SimCC-based (Li et al., 2021c) algorithm that treats keypoint localization as a classification task. Compared with heatmap-based algorithms (Xiao et al., 2018; Zhang et al., 2020; Huang et al., 2020a; Xu et al., 2022b), the SimCC-based algorithm achieves competitive accuracy with lower computational effort. Moreover, SimCC uses a very simple architecture of two fully-connected layers for prediction, making it easy to deploy on various backends.

Fourth, we revisit the training settings in previous works (Bazarevsky et al., 2020; Li et al., 2021a; Lyu et al., 2022), and empirically introduce a collection of training strategies applicable to the pose estimation task. Our experiments demonstrate that this set of strategies bring significant gains to proposed RTMPose as well as other pose estimation models.

Finally, we jointly optimize the inference pipeline. We use the skip-frame detection strategy proposed in Bazarevsky et al. (2020) to reduce the latency and improve the pose-processing with pose Non-Maximum Suppression (NMS) and smoothing filtering for better robustness. In addition, we provide a series of RTMPose models with t/s/m/l/x sizes to cover different application scenarios with the optimum performance-speed trade-off.

We summarize our contributions as follows:

- We design a simple experiment by removing the transposed convolutional layers of SimCC (Li et al., 2021c), revealing that transposed convolutional upsampling layers are redundant (consuming 35% of parameters and 26.7% of computation while only gaining 0.8 AP), which greatly accelerates model inference with minimal performance loss. This finding is encouraging for designing low-computation pose estimation models.

- We outline five aspects influencing multi-person pose estimation performance and latency. We also explore different techniques to improve the performance of the simplified SimCC, providing guidelines and references for designing future industrial-oriented pose estimation algorithms.

- We demonstrate that in scenarios emphasizing lightweight and real-time pose estimation, the top-down paradigm stands out as the optimal choice in terms of both speed and accuracy, breaking the conventional academic stereotype.

- We conduct comprehensive inference speed validation on commonly used deployment frameworks and hardware platforms in the industry. In comparison to currently popular open-source pose estimation pipelines, our proposed RTMPose achieves a superior balance between speed and accuracy across all tests (Fig. 1 Left).

## 2 RELATED WORK

**Bottom-up Approaches.** Bottom-up algorithms (Pishchulin et al., 2016; Cao et al., 2017; Newell et al., 2017; Luo et al., 2021; Cheng et al., 2020; Geng et al., 2021; Kreiss et al., 2019; Jin et al., 2020a) detect instance-agnostic keypoints in an image and partition these keypoints to obtain the human pose. The bottom-up paradigm is considered suitable for crowd scenarios because of the stable computational cost regardless the number of people increases. However, these algorithms often require a large input resolution to handle various person scales, making it challenging to reconcile accuracy and inference speed.

**Top-down Approaches.** Top-down algorithms use off-the-shelf detectors to provide bounding boxes and then crop the human to a uniform scale for pose estimation. Algorithms of the top-down paradigm (Xiao et al., 2018; Cai et al., 2020; Sun et al., 2019; Liu et al., 2021a; Xu et al., 2022b) have been dominating public benchmarks. The two-stage inference paradigm allows both the human detector and the pose estimator to use relatively small input resolutions, which allows them to outperform bottom-up algorithms in terms of speed and accuracy in non-extreme scenarios (i.e. when the number of people in the image is no more than 6). Additionally, most previous work has focused on achieving state-of-the-art performance on public datasets, while our work aims to design models with better speed-accuracy trade-offs to meet the needs of industrial applications.

**Coordinate Classification.** Previous pose estimation approaches usually regard keypoint localization as either coordinate regression (Toshev & Szegedy, 2014; Li et al., 2021a; Mao et al., 2022) or heatmap regression (Xiao et al., 2018; **?**; Zhang et al., 2020; Xu et al., 2022b). SimCC (Li et al., 2021c) introduces a new scheme that formulates keypoint prediction as classification from sub-pixel bins for horizontal and vertical coordinates respectively, which brings about several advantages. First, SimCC is freed from the dependence on high-resolution heatmaps, thus allowing for a very compact architecture that requires neither high-resolution intermediate representations (Sun et al., 2019) nor costly upsampling layers (Xiao et al., 2018). Second, SimCC flattens the final feature map for classification instead of involving global pooling (Toshev & Szegedy, 2014) and therefore avoids the loss of spatial information. Third, the quantization error can be effectively alleviated by coordinate classification at the sub-pixel scale, without the need for extra refinement post-processing (Zhang et al., 2020). These qualities make SimCC attractive for building lightweight pose estimation models. In this work, we further exploit the coordinate classification scheme with optimizations on model architecture and training strategy.

**Vision Transformers.** Transformer-based architectures (Vaswani et al., 2017) ported from modern Natural Language Processing (NLP) have achieved great success in various vision tasks like representation learning (Dosovitskiy et al., 2021; Liu et al., 2021b), object detection (Carion et al., 2020; Zhu et al., 2020; Li et al., 2022), semantic segmentation (Zheng et al., 2021), video understanding (Liu et al., 2022a; Bertasius et al., 2021; Fan et al., 2021), as well as pose estimation (Xu et al., 2022b; Yang et al., 2021; Mao et al., 2022; Li et al., 2021d; Shi et al., 2022). ViTPose (Xu et al., 2022b) leverages the state-of-the-art transformer backbones to boost pose estimation accuracy, while TransPose (Yang et al., 2021) integrates transformer encoders with CNNs to efficiently capture long-range spatial relationships. Token-based keypoint embedding is introduced to incorporate visual cue querying and anatomic constraint learning, shown effective in both heatmap-based (Li et al., 2021d) and regression-based (Mao et al., 2022) approaches. PRTR (Li et al., 2021b) and PETR (Shi et al., 2022) propose end-to-end multi-person pose estimation frameworks with transformers, inspired by the pioneer in detection (Carion et al., 2020). Previous pose estimation approaches with transformers either use a heatmap-based representation or retained both pixel tokens and keypoint tokens, which results in high computation costs and makes real-time inference difficult. In contrast, we incorporate the self-attention mechanism with a compact SimCC-based representation to capture the keypoint dependencies, which significantly reduces the computation load and allows real-time inference with advanced accuracy and efficiency.

## 3 METHODOLOGY

In this section, we expound the roadmap we build RTMPose following the coordinate classification approach. We start by refitting SimCC (Li et al., 2021c) with more efficient backbone architectures,

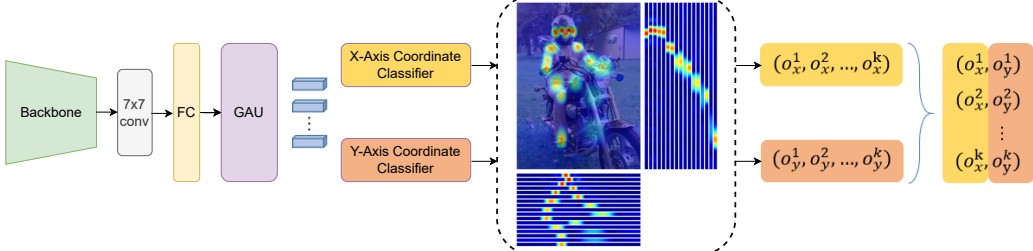

Figure 2: The overall architecture of RTMPose, which contains a convolutional layer, a fully-connected layer and a Gated Attention Unit (GAU) to refine K keypoint representations. After that 2d pose estimation is regarded as two classification tasks for x-axis and y-axis coordinates to predict the horizontal and vertical locations of keypoints.

Table 1: Computational costs and accuracy of baseline methods. We show FLOPs and model parameters of prediction heads for a detailed comparison. "SimCC*" denotes the removal of upsampling layers from the standard SimCC head.

|                 | Heatmap | SimCC   | SimCC*  |
|-----------------|---------|---------|---------|
| Repr. Size      | 64×48   | 512+384 | 512+384 |
| AP              | 71.8    | 72.1    | 71.3    |
| Total FLOPs(G)  | 5.45    | 5.50    | 4.03    |
| Total Params(M) | 34.00   | 36.75   | 23.59   |
| Head FLOPs(G)   | 1.425   | 1.472   | 0.002   |
| Head Params(M)  | 10.492  | 13.245  | 0.079   |

which gives a lightweight yet strong baseline (Sec. 3.1). We adopt the training strategies proposed in RTMDet (Lyu et al., 2022) with minor tweaks to make them more effective on the pose estimation task. The model performance is further improved with a series of delicate modules (Sec. 3.3) and micro designs (Sec. 3.4). Finally, we jointly optimize the entire top-down inference pipeline toward higher speed and better reliability. The final model architecture is shown in Fig. 2, and Fig. 1 Right illustrates the step-by-step gain of the roadmap.

## 3.1 SimCC: A lightweight yet strong baseline

**Preliminary** SimCC (Li et al., 2021c) tackles keypoint localization by treating it as a classification task. It segments horizontal and vertical axes into bins and rounds off coordinates to these bins. The model is trained to predict the bin where a keypoint is. By using a large number of bins, the error is minimized to a subpixel level.

The architecture of SimCCC (Li et al., 2021c) is straightforward, utilizing a single $1 \times 1$ convolution layer for feature extraction and two fully connected layers for classification. It also employs Gaussian label smoothing based on the ground truth, which enhances model performance and aligns with the SORD (Díaz & Marathe, 2019) approach in ordinal regression tasks.

**Baseline** We simplified SimCCC (Li et al., 2021c) by eliminating expensive upsampling layers. As shown in our results in Table 1, this optimized version retains high accuracy while significantly reducing computational costs. By adopting a smaller CSPNext-m (Lyu et al., 2022) backbone, the model becomes more compact, achieving an AP of 69.7%.

## 3.2 Training Techniques

**Pre-training** Previous works (Bazarevsky et al., 2020; Li et al., 2021a) show that pre-training the backbone using the heatmap-based method can improve the model accuracy. We adopt UDP (Huang et al., 2020a) method for the backbone pre-training. This improves the model from 69.7% AP to 70.3% AP. We use this technique as a default setting in the following sections.

**Optimization Strategy**   We adopt the optimization strategy from RTMDet (Lyu et al., 2022). The Exponential Moving Average (EMA) is used for alleviating overfitting (70.3% to 70.4%). The flat cosine annealing strategy improves the accuracy to 70.7% AP. We also inhibit weight decay on normalization layers and biases.

**Two-stage training augmentations**   Following the training strategy in RTMDet (Lyu et al., 2022), we use a strong-then-weak two-stage augmentation. First using strong data augmentations to train 180 epochs and then a weak strategy for 30 epochs. During the strong stage, we use a large random scaling range [0.6, 1.4], and a large random rotation factor, 80, and set the Cutout (DeVries & Taylor, 2017) probability to 1. According to AID (Huang et al., 2020b), Cutout helps to prevent the model from overfitting to the image textures and encourages it to learn the pose structure information. In the weak strategy stage, we turn off the random shift, use a small random rotation range, and set the Cutout probability to 0.5 to allow the model to fine-tune in a domain that more closely matches the real image distribution.

### 3.3   MODULE DESIGN

**Feature dimension**   We observe that the model performance increases along with higher feature resolution. Therefore, we use a fully connected layer to expand the 1D keypoint representations to a desired dimension controlled by the hyper-parameter. In this work, we use 256 dimensions and the accuracy is improved from 71.2% AP to 71.4% AP.

**Self-attention module**   To further exploit the global and local spatial information, we refine the keypoint representations $X$ with a self-attention module, inspired by Li et al. (2021d); Yang et al. (2021). We adopt the transformer variant, Gated Attention Unit (GAU) (Hua et al., 2022), which has faster speed, lower memory cost, and better performance compared to the vanilla transformer (Vaswani et al., 2017). Specifically, GAU improves the Feed-Forward Networks (FFN) in the transformer layer with Gated Linear Unit (GLU) (Shazeer, 2020), and integrates the attention mechanism in an elegant form:

$$
\begin{aligned}
U &= \phi_u(XW_u) \\
V &= \phi_v(XW_v) \\
O &= (U \odot AV)W_o
\end{aligned}
\tag{1}
$$

where $W_u$, $W_v$, and $W_o$ denote the FFNs, and $\odot$ is the pairwise multiplication (Hadamard product) and $\phi$ is the activation function. In this work we implement the self-attention as follows:

$$
A = \frac{1}{n} relu^2 \left( \frac{Q(X)K(Z)^\top}{\sqrt{s}} \right), Z = \phi_z(XW_z)
\tag{2}
$$

where $n$ is the number of input tokens, $W_z$ denotes the FFN, $s = 128$ is the hidden layer dimension, $Q$ and $K$ are simple linear transformations, and $relu^2(\cdot)$ is ReLU then squared. This self-attention module brings about a 0.5% AP (71.9%) improvement to the model performance.

### 3.4   MICRO DESIGN

**Loss function**   We treat the coordinate classification as an ordinal regression task and follow the soft label encoding proposed in SORD (Díaz & Marathe, 2019):

$$
y_i = \frac{e^{\phi(r_t, r_i)}}{\sum_{k=1}^{K} e^{\phi(r_t, r_k)}}
\tag{3}
$$

where $\phi(r_t, r_i)$ is a metric loss function of our choice that penalizes how far the true metric value of $r_t$ is from the rank $r_i \in Y$. In this work, we adopt the unnormalized Gaussian distribution as the inter-class distance metric:

Table 2: Comparison of different kernel sizes

| Kernel Size | mAP |
|---|---|
| 1x1 | 72.8 |
| 3x3 | 73.2 |
| 5x5 | 73.2 |
| **7x7** | **73.3** |
| 9x9 | 73.0 |

Table 3: Comparison of different temperature factors.

| $1/\tau$ | mAP |
|---|---|
| 1 | unstable |
| 5 | 73.1 |
| **10** | **73.3** |
| 15 | 73.0 |

$$\phi(r_t, r_i) = e^{\frac{-(r_t - r_i)^2}{2\sigma^2}} \tag{4}$$

Note that Eq. 3 can be seen as computing Softmax for all $\phi(r_t, r_i)$. We add temperatures in the Softmax operation for both model outputs and soft labels further adjust the normalized distribution shape:

$$y_i = \frac{e^{\phi(r_t, r_i)/\tau}}{\sum_{k=1}^{K} e^{\phi(r_t, r_l)/\tau}} \tag{5}$$

According to the experimental results, using $\tau = 0.1$ can improve the performance from 71.9% to 72.7%.

**Separate $\sigma$**   In SimCCC (Li et al., 2021c), the horizontal and vertical labels are encoded using the same $\sigma$. We empirically explore a simple strategy to set separate $\sigma$ for them: $\sigma = \sqrt{W_S/16}$. $W_S$ is the number of bins in the horizontal and vertical directions respectively. This strategy improves the accuracy from 72.7% to 72.8%.

**Larger convolution kernel & Lower Temperature**   According to Liu et al. (2022b); Yu et al. (2022), the benefits of increasing the kernel size of convolutional layer is significant in Transformer-like architectures. We experiment with different kernel sizes of the last convolutional layer and find that using a larger kernel size gives a performance improvement over using $1 \times 1$ kernel. Finally, we chose to use a $7 \times 7$ convolutional layer, which achieves 73.3% AP. We compare model performances with different kernel sizes in Table 2. Additionally, we also compare the effect of different temperature factors in Table 3 using the final model architecture.

**More epochs and multi-dataset training**   Increasing the training epochs brings extra gains to the model performance. Specifically, training 270 and 420 epochs reach 73.5% AP and 73.7% AP respectively. To further exploit the model's potential, we enrich the training data by combining COCO (Lin et al., 2014) and AI Challenger (Wu et al., 2017) datasets together for pre-training and fine-tuning, with a balanced sampling ratio. The performance finally achieves 75.3% AP.

### 3.5   INFERENCE PIPELINE

Beyond the pose estimation model, we further optimize the overall top-down inference pipeline for lower latency and better robustness. We use the skip-frame detection mechanism as in BlazePose (Bazarevsky et al., 2020), where human detection is performed every K frames, and in the interval frames the bounding boxes are generated from the last pose estimation results. Additionally, to achieve smooth prediction over frames, we use OKS-based pose NMS and OneEuro (Casiez et al., 2012) filter in the post-processing stage.

## 4   EXPERIMENTS

### 4.1   SETTINGS

We trained the RTMPose models using the AdamW optimizer with a base learning rate of 0.004. The learning rate followed a Flat-Cosine schedule. Training used a batch size of 1024 and included

Table 4: Body pose estimation results on COCO validation set. We only report GFLOPs of pose model and ignore the detection model. Flip test is not used.

| Methods | | Backbone | Detector | Det. Input Size | Pose Input Size | GFLOPs | AP | Extra Data |
|---|---|---|---|---|---|---|---|---|
| PaddleDetection (Authors) | TinyPose | Wider NLiteHRNet | YOLOv3 | $608 \times 608$ | $128 \times 96$ | 0.08 | 52.3 | |
| | TinyPose | Wider NLiteHRNet | YOLOv3 | $608 \times 608$ | $256 \times 192$ | 0.33 | 60.9 | |
| | TinyPose | Wider NLiteHRNet | Faster-RCNN | N/A | $128 \times 96$ | 0.08 | 56.1 | AIC(220K) |
| | TinyPose | Wider NLiteHRNet | Faster-RCNN | N/A | $256 \times 192$ | 0.33 | 65.6 | +Internal(unknown) |
| | TinyPose | Wider NLiteHRNet | PicoDet-s | $320 \times 320$ | $128 \times 96$ | 0.08 | 48.4 | |
| | TinyPose | Wider NLiteHRNet | PicoDet-s | $320 \times 320$ | $256 \times 192$ | 0.33 | 56.5 | |
| AlphaPose (Fang et al., 2022) | FastPose | ResNet 50 | YOLOv3 | $608 \times 608$ | $256 \times 192$ | 5.91 | 71.2 | |
| | FastPose(DUC) | ResNet-50 | YOLOv3 | $608 \times 608$ | $256 \times 192$ | 9.71 | 71.7 | |
| | FastPose(DUC) | ResNet-152 | YOLOv3 | $608 \times 608$ | $256 \times 192$ | 15.99 | 72.6 | |
| | FastPose | ResNet 50 | Faster-RCNN | N/A | $256 \times 192$ | 5.91 | 69.7 | - |
| | FastPose(DUC) | ResNet-50 | Faster-RCNN | N/A | $256 \times 192$ | 9.71 | 70.3 | |
| | FastPose(DUC) | ResNet-152 | Faster-RCNN | N/A | $256 \times 192$ | 15.99 | 71.3 | |
| MMPose (Contributors, 2020) | RTMPose-t | CSPNeXt-t | Faster-RCNN | N/A | $256 \times 192$ | 0.36 | 65.8 | |
| | RTMPose-s | CSPNeXt-s | Faster-RCNN | N/A | $256 \times 192$ | 0.68 | 69.6 | |
| | RTMPose-m | CSPNeXt-m | Faster-RCNN | N/A | $256 \times 192$ | 1.93 | 73.6 | - |
| | RTMPose-l | CSPNeXt-l | Faster-RCNN | N/A | $256 \times 192$ | 4.16 | 74.8 | |
| | RTMPose-t | CSPNeXt-t | YOLOv3 | $608 \times 608$ | $256 \times 192$ | 0.36 | 66.0 | |
| | RTMPose-s | CSPNeXt-s | YOLOv3 | $608 \times 608$ | $256 \times 192$ | 0.68 | 70.3 | |
| | RTMPose-m | CSPNeXt-m | YOLOv3 | $608 \times 608$ | $256 \times 192$ | 1.93 | 74.7 | |
| | RTMPose-l | CSPNeXt-l | YOLOv3 | $608 \times 608$ | $256 \times 192$ | 4.16 | 75.7 | |
| | RTMPose-t | CSPNeXt-t | Faster-RCNN | N/A | $256 \times 192$ | 0.36 | 67.1 | |
| | RTMPose-s | CSPNeXt-s | Faster-RCNN | N/A | $256 \times 192$ | 0.68 | 71.1 | |
| | RTMPose-m | CSPNeXt-m | Faster-RCNN | N/A | $256 \times 192$ | 1.93 | 75.3 | |
| | RTMPose-l | CSPNeXt-l | Faster-RCNN | N/A | $256 \times 192$ | 4.16 | 76.3 | AIC(220K) |
| | RTMPose-t | CSPNeXt-t | PicoDet-s | $320 \times 320$ | $256 \times 192$ | 0.36 | 64.3 | |
| | RTMPose-s | CSPNeXt-s | PicoDet-s | $320 \times 320$ | $256 \times 192$ | 0.68 | 68.8 | |
| | RTMPose-m | CSPNeXt-m | PicoDet-s | $320 \times 320$ | $256 \times 192$ | 1.93 | 73.2 | |
| | RTMPose-l | CSPNeXt-l | PicoDet-s | $320 \times 320$ | $256 \times 192$ | 4.16 | 74.2 | |
| | RTMPose-t | CSPNeXt-t | RTMDet-nano | $320 \times 320$ | $256 \times 192$ | 0.36 | 64.4 | |
| | RTMPose-s | CSPNeXt-s | RTMDet-nano | $320 \times 320$ | $256 \times 192$ | 0.68 | 68.5 | |
| | RTMPose-m | CSPNeXt-m | RTMDet-nano | $320 \times 320$ | $256 \times 192$ | 1.93 | 73.2 | |
| | RTMPose-l | CSPNeXt-l | RTMDet-nano | $320 \times 320$ | $256 \times 192$ | 4.16 | 74.2 | |
| | RTMPose-m | CSPNeXt-m | RTMDet-m | $640 \times 640$ | $256 \times 192$ | 1.93 | 75.7 | |
| | RTMPose-l | CSPNeXt-l | RTMDet-m | $640 \times 640$ | $256 \times 192$ | 4.16 | 76.6 | |

1000 warm-up iterations. Weight decay was set at 0.05 for RTMPose-m/l and zero for RTMPose-t/s. We employed an Exponential Moving Average (EMA) decay of 0.9998 for RTMPose-s/m/l/x, while RTMPose-t did not utilize EMA. The training epochs consisted of 210 for pre-training and 420 for fine-tuning. As described in Sec. 3.2, we conduct a heatmap-based pre-training (Huang et al., 2020a) which follows the same training strategies used in the fine-tuning except for shorter epochs. All our models are trained on 8 NVIDIA A100 GPUs. And we evaluate the model performance by mean Average Precision (AP).

Table 5: Whole-body pose estimation results on COCO-WholeBody (Jin et al., 2020b; Xu et al., 2022a) V1.0 dataset. We only report the input size and GFLOPs of pose models in top-down approaches and ignore the detection model. "*" denotes the model is pre-trained on AIC+COCO. "†" indicates multi-scale testing. Flip test is used.

| | Method | Input Size | GFLOPs | whole-body | | body | | foot | | face | | hand | |
|---|---|---|---|---|---|---|---|---|---|---|---|---|---|
| | | | | AP | AR | AP | AR | AP | AR | AP | AR | AP | AR |
| Whole-body | SN† (Hidalgo et al., 2019) | N/A | 272.3 | 32.7 | 45.6 | 42.7 | 58.3 | 9.9 | 36.9 | 64.9 | 69.7 | 40.8 | 58.0 |
| | OpenPose (Cao et al., 2019) | N/A | 451.1 | 44.2 | 52.3 | 56.3 | 61.2 | 53.2 | 64.5 | 76.5 | 84.0 | 38.6 | 43.3 |
| Bottom-up | PAF† (Cao et al., 2017) | 512×512 | 329.1 | 29.5 | 40.5 | 38.1 | 52.6 | 5.3 | 27.8 | 65.6 | 70.1 | 35.9 | 52.8 |
| | AE (Newell et al., 2017) | 512×512 | 212.4 | 44.0 | 54.5 | 58.0 | 66.1 | 57.7 | 72.5 | 58.8 | 65.4 | 48.1 | 57.4 |
| Top-down | DeepPose (Toshev & Szegedy, 2014) | 384×288 | 17.3 | 33.5 | 48.4 | 44.4 | 56.8 | 36.8 | 53.7 | 49.3 | 66.3 | 23.5 | 41.0 |
| | SimpleBaseline (Xiao et al., 2018) | 384×288 | 20.4 | 57.3 | 67.1 | 66.6 | 74.7 | 63.5 | 76.3 | 73.2 | 81.2 | 53.7 | 64.7 |
| | HRNet (Sun et al., 2019) | 384×288 | 16.0 | 58.6 | 67.4 | 70.1 | 77.3 | 58.6 | 69.2 | 72.7 | 78.3 | 51.6 | 60.4 |
| | PVT (Wang et al., 2021) | 384×288 | 19.7 | 58.9 | 68.9 | 67.3 | 76.1 | 66.0 | 79.4 | 74.5 | 82.2 | 54.5 | 65.4 |
| | FastPose50-dcn-si (Fang et al., 2022) | 256×192 | 6.1 | 59.2 | 66.5 | 70.6 | 75.6 | 70.2 | 77.5 | 77.5 | 82.5 | 45.7 | 53.9 |
| | ZoomNet (Jin et al., 2020b) | 384×288 | 28.5 | 63.0 | 74.2 | **74.5** | **81.0** | 60.9 | 70.8 | 88.0 | 92.4 | 57.9 | 73.4 |
| | ZoomNAS (Xu et al., 2022a) | 384×288 | 18.0 | **65.4** | **74.4** | 74.0 | 80.7 | 61.7 | 71.8 | 88.9 | **93.0** | **62.5** | **74.0** |
| | RTMPose-m* | 256×192 | 2.2 | 58.2 | 67.4 | 67.3 | 75.0 | 61.5 | 75.2 | 81.3 | 87.1 | 47.5 | 58.9 |
| | RTMPose-l* | 256×192 | 4.5 | 61.1 | 70.0 | 69.5 | 76.9 | 65.8 | 78.5 | 83.3 | 88.7 | 51.9 | 62.8 |
| | RTMPose-l* | 384×288 | 10.1 | 64.8 | 73.0 | 71.2 | 78.1 | **69.3** | **81.1** | 88.2 | 91.9 | 57.9 | 67.7 |
| | RTMPose-x | 384×288 | 18.1 | 65.2 | 73.2 | 71.2 | 78.0 | 68.1 | 80.4 | **89.0** | 92.2 | 59.3 | 68.7 |
| | RTMPose-x* | 384×288 | 18.1 | 65.3 | 73.3 | 71.4 | 78.4 | 69.2 | 81.0 | 88.9 | 92.3 | 59.0 | 68.5 |

## 4.2 BENCHMARK RESULTS

**COCO**    COCO (Lin et al., 2014) is the most popular benchmark for 2d body pose estimation. We follow the standard splitting of `train2017` and `val2017`, which contains 118K and 5k images for training and validation respectively. We extensively study the pose estimation performance with different off-the-shelf detectors including YOLOv3 (Redmon & Farhadi, 2018), Faster-RCNN (Ren et al., 2015), and RTMDet (Lyu et al., 2022). To conduct a fair comparison with AlphaPose (Fang et al., 2022) which doesn't use extra training data, we also report the performance of RTMPose only trained on COCO. As shown in Table 4, RTMPose outperforms competitors by a large margin with much lower complexity and shows strong robustness for detection.

**COCO-SinglePerson**    For a fair comparison with popular pose estimation open-source algorithms like BlazePose (Bazarevsky et al., 2020), MoveNet (Votel et al., 2023), and PaddleDetection (Authors), we construct a COCO-SinglePerson dataset that contains 1045 single-person images from the COCO `val2017` set to evaluate RTMPose as well as other approaches. The evaluation results can be found in Appendix A.1.

**COCO-WholeBody**    We also validate the proposed RTMPose model on the whole-body pose estimation task with COCO-WholeBody (Jin et al., 2020b; Xu et al., 2022a) V1.0 dataset. As shown in Table 5, RTMPose achieves superior performance and well balances accuracy and complexity. Specifically, our RTMPose-m model outperforms previous open-source libraries (Cao et al., 2019; Fang et al., 2022; Yu et al., 2021a) with significantly lower GFLOPs. And by increasing the input resolution and training data we obtain competitive accuracy with SOTA approaches (Jin et al., 2020b; Xu et al., 2022a).

**Other Datasets**    As shown in Table 6 and Table 7, we further evaluate RTMPose on AP-10K (Yu et al., 2021b), CrowdPose (Li et al., 2018) and MPII (Andriluka et al., 2014) datasets. We report the model performance using ImageNet (Deng et al., 2009) pre-training for a fair comparison with baselines. Besides we also report the performance of our models pre-trained using a combination of COCO (Lin et al., 2014) and AI Challenger (AIC) (Wu et al., 2017), which achieves higher accuracy and can be easily reproduced by users with our provided pre-trained weights.

Table 6: Performance on different datasets. "*" denotes the model is pre-trained on AIC+COCO and fine-tuned on the corresponding dataset. Flip test is used.

| Dataset | Methods | Backbone | Input Size | GFLOPs | AP |
|---|---|---|---|---|---|
| AP-10K (Yu et al., 2021b) | SimpleBaseline (Xiao et al., 2018) | ResNet-50 | $256 \times 256$ | 7.28 | 68.0 |
| | HRNet (Sun et al., 2019) | HRNet-w32 | $256 \times 256$ | 10.27 | 72.2 |
| | RTMPose-m | CSPNeXt-m | $256 \times 256$ | **2.57** | **68.4** |
| | RTMPose-m* | CSPNeXt-m | $256 \times 256$ | **2.57** | **72.2** |
| CrowdPose (Li et al., 2018) | SimpleBaseline (Xiao et al., 2018) | ResNet-50 | $256 \times 192$ | 5.46 | 63.7 |
| | HRNet (Sun et al., 2019) | HRNet-w32 | $256 \times 192$ | 7.7 | 67.5 |
| | RTMPose-m | CSPNeXt-m | $256 \times 192$ | **1.93** | **66.9** |
| | RTMPose-m* | CSPNeXt-m | $256 \times 192$ | **1.93** | **70.6** |

## 4.3 INFERENCE SPEED

We perform the export, deployment, inference, and testing of models by MMDeploy (Contributors, 2021) to test the inference speed on CPU and GPU respectively. To align with industrial deployment scenarios, we adopt less performant but more commonly used devices in the industry. The TensorRT inference latency is tested in the half-precision floating-point format (FP16) on an NVIDIA GeForce GTX 1660 Ti GPU, and the ONNX latency is tested on an Intel I7-11700 CPU with ONNXRuntime with 1 thread. The inference batch size is 1. All models are tested on the same devices with 50 times warmup and 200 times inference for fair comparison. The results are shown in Table 8.

Furthermore, we examined the inference speed of RTMPose on a mobile device, specifically the Snapdragon 865 chip. RTMPose-t achieved a frame speed of 9.02 ms, which is comparable to

TinyPose (Authors). However, it's important to note that RTMPose offers a wider selection of high-accuracy models to meet more realistic requirements. We also assessed the pipeline speed of RTMPose, with RTMPose-s processing frames in 21.7 ms while achieving an average precision (AP) of 68.5 on COCO-val. For additional details, please refer to the Appendix A.2.

Table 7: Comparison on MPII (Andriluka et al., 2014) validation set. "*" denotes the model is pre-trained on AIC+COCO and fine-tuned on MPII. Flip test is used.

| Dataset | Methods | Backbone | Input Size | GFLOPs | PCKh@0.5 |
|---------|---------|----------|-----------|--------|----------|
| MPII (Andriluka et al., 2014) | SimpleBaseline (Xiao et al., 2018) | ResNet-50 | $256 \times 256$ | 7.28 | 88.2 |
| | HRNet (Sun et al., 2019) | HRNet-w32 | $256 \times 256$ | 10.27 | 90.0 |
| | SimCC (Li et al., 2021c) | HRNet-w32 | $256 \times 256$ | 10.34 | 90.0 |
| | TokenPose (Li et al., 2021d) | L/D24 | $256 \times 256$ | 11.0 | 90.2 |
| | RTMPose-m | CSPNeXt-m | $256 \times 256$ | **2.57** | **88.9** |
| | RTMPose-m* | CSPNeXt-m | $256 \times 256$ | **2.57** | **90.7** |

Table 8: Inference speed on CPU and GPU. RTMPose models are deployed and tested using ON-NXRuntime and TensorRT respectively. Flip test is not used in this table.

| Results | | Input Size | GFLOPs | AP | CPU(ms) | GPU(ms) |
|---------|---|-----------|--------|-----|---------|---------|
| COCO (Lin et al., 2014) | TinyPose | $256 \times 192$ | 0.33 | 65.6 | 10.580 | 3.055 |
| | LiteHRNet-30 | $256 \times 192$ | 0.42 | 66.3 | 22.750 | 6.561 |
| | RTMPose-t | $256 \times 192$ | **0.36** | **67.1** | **3.204** | **1.064** |
| | RTMPose-s | $256 \times 192$ | **0.68** | **71.2** | **4.481** | **1.392** |
| | HRNet-w32+UDP | $256 \times 192$ | 7.7 | 75.1 | 37.734 | 5.133 |
| | RTMPose-m | $256 \times 192$ | **1.93** | **75.3** | **11.060** | **2.288** |
| | RTMPose-l | $256 \times 192$ | **4.16** | **76.3** | **18.847** | **3.459** |
| COCO-WholeBody (Jin et al., 2020b) | HRNet-w32+DARK | $256 \times 192$ | 7.72 | 57.8 | 39.051 | 5.154 |
| | RTMPose-m | $256 \times 192$ | **2.22** | **59.1** | **13.496** | **4.000** |
| | RTMPose-l | $256 \times 192$ | **4.52** | **62.2** | **23.410** | **5.673** |
| | HRNet-w48+DARK | $384 \times 288$ | 35.52 | 65.3 | 150.765 | 13.974 |
| | RTMPose-l | $384 \times 288$ | **10.07** | **66.1** | **44.581** | **7.678** |

## 5 CONCLUSION

This paper empirically explores five key factors in pose estimation the pipeline such as the paradigm, model architecture, training strategy, and deployment. Based on the findings we present a high-performance real-time multi-person pose estimation pipeline, RTMPose, which achieves excellence in balancing model performance and complexity and can be deployed on various devices (CPU, GPU, and mobile devices) for real-time inference. We hope that the proposed algorithm alone with its open-sourced implementation can meet some of the demand for applicable pose estimation in industry, and benefit future explorations on the human pose estimation task.

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

# A APPENDIX

## A.1 MORE BENCHMARK RESULTS

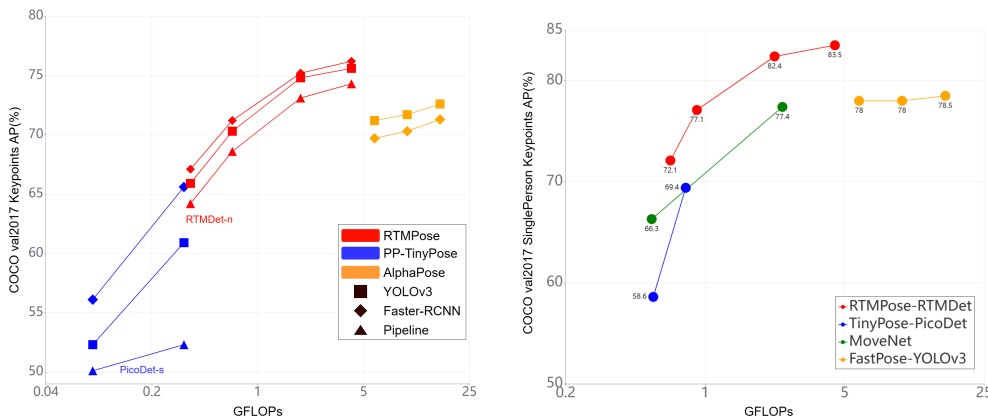

Figure 3: Comparison of GFLOPs and accuracy. Left: Comparison of RTMPose and other open-source pose estimation libraries on full COCO val set. Right: Comparison of RTMPose and other open-source pose estimation libraries on COCO-SinglePerson val set.

**COCO-SinglePerson** Popular pose estimation open-source libraries like BlazePose (Bazarevsky et al., 2020), MoveNet (Votel et al., 2023), and PaddleDetection (Authors) are designed primarily for single-person or sparse scenarios, which are practical in mobile applications and human-machine interactions. For a fair comparison, we construct a COCO-SinglePerson dataset that contains 1045 single-person images from the COCO `val2017` set to evaluate RTMPose as well as other approaches. For MoveNet (Votel et al., 2023), we follow the official inference pipeline to apply a cropping algorithm, namely using the coarse pose prediction of the first inference to crop the input image and performing a second inference for better pose estimation results. The evaluation results in Table 9 and Fig. 3 show that RTMPose archives superior performance and efficiency even compared to previous solutions tailored for the single-person scenario.

Table 9: Body pose estimation results on COCO-SinglePerson validation set. We sum up top-down methods' GFLOPs of detection and pose for a fair comparison with bottom-up methods. "*" denotes double inference. Flip test is not used.

| Methods | | Backbone | Detector | Det. Input Size | Pose Input Size | GFLOPs | AP | Extra Data |
|---|---|---|---|---|---|---|---|---|
| MediaPipe (Bazarevsky et al., 2020) | BlazePose-Lite | BlazePose | N/A | $256 \times 256$ | N/A | N/A | 29.3 | Internal(85K) |
| | BlazePose-Full | BlazePose | N/A | $256 \times 256$ | N/A | N/A | 35.4 | |
| MoveNet (Votel et al., 2023) | Lightning | MobileNetv2 | N/A | $192 \times 192$ | N/A | 0.54 | 53.6* | Internal(23.5K) |
| | Thunder | MobileNetv2 depth×1.75 | N/A | $256 \times 256$ | N/A | 2.44 | 64.8* | |
| PaddleDetection (Authors) | TinyPose | Wider NLiteHRNet | PicoDet-s | $320 \times 320$ | $128 \times 96$ | 0.55 | 58.6 | AIC(220K) |
| | TinyPose | Wider NLiteHRNet | PicoDet-s | $320 \times 320$ | $256 \times 192$ | 0.80 | 69.4 | +Internal(unknown) |
| MMPose (Contributors, 2020) | RTMPose-t | CSPNeXt-t | RTMDet-nano | $320 \times 320$ | $256 \times 192$ | 0.67 | 72.1 | AIC(220K) |
| | RTMPose-s | CSPNeXt-s | RTMDet-nano | $320 \times 320$ | $256 \times 192$ | 0.91 | 77.1 | |
| | RTMPose-m | CSPNeXt-m | RTMDet-nano | $320 \times 320$ | $256 \times 192$ | 2.23 | 82.4 | |
| | RTMPose-l | CSPNeXt-l | RTMDet-nano | $320 \times 320$ | $256 \times 192$ | 4.47 | 83.5 | |

## A.2 INFERENCE SPEED

In this appendix, we extend our experimentation to assess the inference speed of RTMPose on a mobile device using ncnn for deployment and testing. Table 10 demonstrates the comparison of inference speed on the mobile device, specifically the Snapdragon 865 chip with RTMPose models.

Furthermore, we maintained our evaluation of TensorRT inference latency on an NVIDIA GeForce GTX 1660 Ti GPU in the half-precision floating-point format (FP16) and ONNX latency on an Intel I7-11700 CPU with ONNXRuntime, using a single thread. The inference batch size remained

consistent at 1. All models underwent a rigorous testing regimen on the same devices, including 50 warm-up runs and 200 inference runs to ensure a fair comparison.

For a comprehensive evaluation, we also included TinyPose (Authors) in our tests, assessing it with both MMDeploy and FastDeploy. We observed that ONNXRuntime speed on MMDeploy was slightly faster (10.58 ms vs. 12.84 ms). The detailed results can be found in Table 10.

Table 10: Comparison of inference speed on Snapdragon 865. RTMPose models are deployed and tested using ncnn.

| Methods | | Input Size | GFLOPs | AP(GT) | FP32(ms) | FP16(ms) |
|---|---|---|---|---|---|---|
| PaddleDetection (Authors) | TinyPose | $128 \times 96$ | 0.08 | 58.4 | 4.57 | 3.27 |
| | TinyPose | $256 \times 192$ | 0.33 | 68.3 | 14.07 | 8.33 |
| MMPose (Contributors, 2020) | RTMPose-t | $256 \times 192$ | 0.36 | 68.4 | 15.84 | 9.02 |
| | RTMPose-s | $256 \times 192$ | 0.68 | 72.8 | 25.01 | 13.89 |
| | RTMPose-m | $256 \times 192$ | 1.93 | 77.3 | 49.46 | 26.44 |
| | RTMPose-l | $256 \times 192$ | 4.16 | 78.3 | 85.75 | 45.37 |

Table 11 analyzes inference speeds across models and devices, revealing the balance between accuracy and speed. RTMPose performs well across sizes, while RTMDet-nano prioritizes efficiency. This data aids in selecting models for diverse real-time applications.

Table 11: Pipeline Inference speed on CPU, GPU and Mobile device.

| Model | Input Size | GFLOPs | Pipeline AP | CPU(ms) | GPU(ms) | Mobile(ms) |
|---|---|---|---|---|---|---|
| RTMDet-nano | $320 \times 320$ | 0.31 | 64.4 | 12.403 | 2.467 | 18.780 |
| RTMPose-t | $256 \times 192$ | 0.36 | | | | |
| RTMDet-nano | $320 \times 320$ | 0.31 | 68.5 | 16.658 | 2.730 | 21.683 |
| RTMPose-s | $256 \times 192$ | 0.42 | | | | |
| RTMDet-nano | $320 \times 320$ | 0.31 | 73.2 | 26.613 | 4.312 | 32.122 |
| RTMPose-m | $256 \times 192$ | 1.93 | | | | |
| RTMDet-nano | $320 \times 320$ | 0.31 | 74.2 | 36.311 | 4.644 | 47.642 |
| RTMPose-l | $256 \times 192$ | 4.16 | | | | |

