# OpenReview forum: "RTMPose: Real-Time Models for Multi-Person Pose Estimation"
_ICLR.cc/2024/Conference — Submitted to ICLR 2024_

### Official Review · Reviewer_hfGj · 2023-10-30

**Soundness:** 4 excellent
**Presentation:** 3 good
**Contribution:** 3 good
**Rating:** 8
**Confidence:** 4

**Summary:**

The authors present a high-performance real-time multi-person pose estimation model, which can achieve real-time inference speed on CPU, GPU, and mobile devices. This article may provide guidelines and references for designing future industrial-oriented pose estimation algorithms.

**Strengths:**

- The paper is well-written and easy to follow. The authors provide clear explanations of the paradigm, backbone network, localization method, training strategy, and deployment.
- The paper conducts comprehensive inference speed validation on commonly used deployment frameworks and hardware platforms in the industry.
- The paper also includes helpful visualizations and figures to illustrate the key concepts.

**Weaknesses:**

- Table 4 has a lot of content, but the analysis of the results is very thin.
- The author did not analyze why Large Kernel Convolution works. Some heat maps may be helpful for analysis.
- The author did not analyze why NVIDIA GeForce GTX 1660 Ti GPU and Intel I7-11700 CPU were chosen. Has the author tried other devices?

**Questions:**

Please refer to the Weakness above.

**Details Of Ethics Concerns:**

None.

---

> ### Author Response · Authors · 2023-11-21
>
> Dear Reviewer,
>
> Thank you for your thorough review and constructive feedback on our manuscript. We appreciate your positive evaluation and have taken note of the specific points raised in your review.
>
> 1. **Large Kernel Convolution:**
>    We want to express our gratitude for your insightful comments regarding the Large Kernel Convolution. The inspiration for incorporating this technique came from a series of works, including MetaFormer and ConvNext, which demonstrated the effectiveness of enlarging convolutional kernel sizes within transformer-like structures. This modification is part of our efforts to enhance the performance of Simplified SimCC, and our experiments have indeed validated its efficacy.
>
> 2. **Heatmap Visualization:**
>    Regarding your suggestion on using heatmaps for analysis, we appreciate the suggestion. In fact, we did explore heatmap visualization in our experiments. However, it's important to note that SimCC differs from traditional heatmap-based methods, and in the case of Large Kernel Convolution, the feature maps received are not typical heatmaps. The convolutional layer here functions more like a spatial token mixer, making the visualization less interpretable compared to heatmap-based methods.
>
> 3. **Device Selection:**
>    We appreciate your inquiry regarding the choice of devices. Our decision to use the NVIDIA GeForce GTX 1660 Ti GPU and Intel I7-11700 CPU was deliberate. Given the focus of our work on designing a lightweight pose estimation model for common industrial scenarios, we opted for lower-cost, industry-common devices. This choice ensures that our results are directly applicable and valuable for industrial deployment.
>
> We are grateful for your positive rating and the confidence you have expressed in our work. Your thoughtful comments have provided valuable insights, and we believe that the clarifications and additional information provided address the concerns raised.
>
> Thank you for your time and consideration.

---

### Official Review · Reviewer_Snzb · 2023-10-30

**Soundness:** 3 good
**Presentation:** 3 good
**Contribution:** 2 fair
**Rating:** 5
**Confidence:** 4

**Summary:**

This paper aims at real-time multi-person pose estimation. It empirically explores key factors in pose estimation including paradigm, model architecture, training strategy, and deployment, and presents a high-performance, real-time multi-person pose estimation pipeline. Experimental results show that the proposed method achieves an excellent balance between performance and complexity. It can also be deployed on various devices (CPU, GPU, and mobile devices) for real-time inference.

**Strengths:**

-The proposed method empirically integrates key modules or factors in existing methods that contribute to real-time pose estimation, and an ablation study of each improving factor is given.

-The experimental results are impressive, demonstrating the high performance and efficiency of the proposed method.

**Weaknesses:**

1. Despite its high performance and efficiency, the proposed method is an integrated engineering framework of existing methods and training tricks, lacks its original methodological contributions, and is not suitable for top academic conferences like ICLR.

2. The writing can be improved. For example, 1) some symbols in Equations 1 and 2 are not defined, which should not be ignored for an academic paper; 2) some references have no journal or conference information (e.g., Huang 2020c, Li 2021c, Lyu 2022, etc.).

3. Typos. On page 4, "Table 3.1" should be "Table 1".

**Questions:**

Please see the weaknesses.

---

> ### Author Response · Authors · 2023-11-21
>
> Dear Reviewer,
>
> Thank you for your thoughtful evaluation of our manuscript. We appreciate your constructive feedback, and we have taken several steps to address the concerns raised in your review.
>
> 1. **Presentation Quality:**
>    We have thoroughly reviewed the presentation of the paper and addressed the identified issues, including typos and repetitive references. Notably, we have supplemented the definitions of symbols in Equations 1 and 2 to ensure clarity and precision in the academic context.
>
> 2. **Contribution Clarification:**
>    - Our work originated from the identification of significant redundancies within the SimCC method (consuming 35\% of parameters and 26.7\% of computation while only gaining 0.8 AP). We believe that the simplification of the structure, as explored in our research, holds immense potential in the domain of lightweight pose estimation. Subsequently, we conducted a systematic analysis of five factors influencing real-time multi-person pose estimation and designed a comprehensive real-time pose estimation pipeline based on these insights.
>    - While we acknowledge that our proposed method integrates existing modules and training strategies, we believe we are the first to comprehensively address these factors in a single study, making a systematic contribution to the field of lightweight pose estimation.
>    - An important contribution of our work is challenging the prevailing stereotype in the academic community regarding top-down algorithmsWe understand that our innovation is more methodological than algorithmic, and we hope this perspective aligns with your expectations.
>
> We appreciate your detailed feedback and are confident that these revisions significantly enhance the clarity and impact of our contributions. We look forward to any further guidance or suggestions you may have.
>
> Thank you for your time and consideration.

---

> > ### Comment · Reviewer_Snzb · 2023-11-23
> > **Responses to official comment by authors**
> >
> > Thank the authors for the responses. However, my main concern regarding limited original methodological contributions is not solved. I would suggest the authors consider other engineering conferences.

---

### Official Review · Reviewer_KXJ9 · 2023-11-01

**Soundness:** 3 good
**Presentation:** 1 poor
**Contribution:** 2 fair
**Rating:** 3
**Confidence:** 5

**Summary:**

This paper presents RTMPose, which is fast on mobile device and accurate at the same time. It explores five influencing factors to the performance and latency of multi-person pose estimation. By exploring the factors, RTMPose have a good balance between speed and performance.

**Strengths:**

1. Experiments are extensive. The five factors are thoroughly discussed and verified.
2. Results achieve nice balance between speed and performance, also on mobile devices.

**Weaknesses:**

1. The presentation of the paper is not good. There are a lot of typos and repetitive references. e.g., UDP (Huang et al., 2020), Crowdpose (Li et al., 2018);  Typos: Table 3.1?
2. Although experiments are extensively done, there is no interesting insight into the five factors, which seems hyper-parameter tuning to me.
In general, I don't think this paper is ready to be accepted.

**Questions:**

Please see weaknesses

---

> ### Author Response · Authors · 2023-11-21
>
> Dear Reviewer,
>
> Thank you for your detailed evaluation of our manuscript. We appreciate your constructive feedback and have carefully addressed the concerns raised in your review.
>
> 1. **Presentation Quality**:
>     We have thoroughly reviewed the presentation of the paper and addressed the identified issues, including typos and repetitive references.
>
> 2. **Contribution Refinement**:
>     In response to your comments on the contribution section, we have refined our statements to better highlight the unique aspects of our work:
>     - We have conducted simple yet insightful experiments to demonstrate the presence of redundant components in the SimCC method (consuming 35\% of parameters and 26.7\% of computation while only gaining 0.8 AP). The identification of these redundancies underscores the substantial potential for a simplified structure in the domain of lightweight pose estimation.
>     - An important contribution of our work is challenging the prevailing stereotype in the academic community regarding top-down algorithms. Contrary to the common belief that top-down algorithms sacrifice speed for accuracy, our research establishes that top-down approaches can achieve both high speed and precision, making them an optimal choice for industrial pose estimation algorithm design.
>     - Our paper provides a systematic analysis of the five aspects influencing real-time multi-person pose estimation. Additionally, we have designed a comprehensive real-time pose estimation pipeline based on this analysis.
>
> We appreciate your insights and believe that these revisions significantly enhance the clarity and impact of our contributions. We are confident that the revised manuscript will address your concerns and provide a more compelling case for acceptance.
>
> Thank you for your time and consideration.

---

### Meta-Review · Area_Chair_DPWH · 2023-12-09

**Metareview:**

This paper tackles multi-person pose estimation and explores five different factors which influence run-time speed for real-time pose estimation.  The main strength is the extensive experimentation and the very fast run-time speeds achieved.  However, two of the three reviewers cite concerns regarding the lack of methodological contributions and insight into the performance, especially for a conference like ICLR.  The AC concurs with this decision and recommends that the authors submit to a more application oriented venue.

**Justification For Why Not Higher Score:**

lack of methodological contribution and insight.

**Justification For Why Not Lower Score:**

N/A

---

### Decision · Program_Chairs · 2024-01-16

Reject